# Rolling-Circle-Amplification-Assisted DNA Biosensors for Sensitive and Specific Detection of Hypochlorous Acid and Myeloperoxidase

Bo Liu [1,†], Jia-Yi Ma [1,†], Jing Wang [2], Dong-Xia Wang [1], An-Na Tang [1] and De-Ming Kong [1,*]

1   State Key Laboratory of Medicinal Chemical Biology, Tianjin Key Laboratory of Biosensing and Molecular Recognition, Research Centre for Analytical Sciences, College of Chemistry, Nankai University, Tianjin 300071, China; nankai_lb@163.com (B.L.); 2120170751@mail.nankai.edu.cn (D.-X.W.); tanganna@nankai.edu.cn (A.-N.T.)
2   School of Medical Laboratory, College of Medical Technology, Tianjin Medical University, Guangdong Road, Tianjin 300203, China; wj822@mail.nankai.edu.cn
*   Correspondence: kongdem@nankai.edu.cn
†   These authors contributed equally to this work.

**Abstract:** Hypochlorous acid (HClO) is a common reactive oxygen species (ROS), with a high chemical reactivity. Myeloperoxidase (MPO) is an enzyme that catalyzes in vivo redox reactions between $H_2O_2$ and $Cl^-$ to produce HClO. Abnormal levels of HClO and MPO may lead to oxidative stress, irreversible tissue damage and, thus, serious diseases; they are thus becoming important biomarkers and therapeutic targets. In this work, using HClO-induced site-specific cleavage of phosphorothioate-modified DNA to trigger rolling circle amplification (RCA), RCA-assisted biosensors have been developed for the highly sensitive and specific detection of HClO and MPO. Only two DNA oligonucleotides are used in the sensing systems. The powerful signal-amplification capability of RCA endows the sensing systems with a high sensitivity, and the specific fluorescent response of thioflavin T (ThT) to G-quadruplexes in RCA products makes a label-free signal output possible. The proposed biosensors were demonstrated to work well not only for the sensitive and specific quantitation of HClO and MPO with detection limits of 1.67 nM and 0.33 ng/mL, respectively, but also for the screening and inhibitory capacity evaluation of MPO inhibitors, thus holding great promise in disease diagnosis and drug analysis.

**Keywords:** hypochlorous acid; myeloperoxidase; rolling circle amplification; phosphorothioate; biosensor

## 1. Introduction

Reactive oxygen species (ROS) are a kind of oxygen-containing substance with high reactivity, playing vital roles in multiple physiological and pathological processes [1,2]. Hypochlorous acid (HClO) is an important member of the ROS family. Under physiological conditions, HClO can be produced through the reaction of hydrogen peroxide ($H_2O_2$) and chloride ion ($Cl^-$) catalyzed by myeloperoxidase (MPO) [3]. Therefore, there is a high correlation between HClO and MPO in organisms. Abnormal levels of HClO and MPO may lead to oxidative stress, irreversible tissue damage and, thus, serious diseases, including inflammatory reactions, atherosclerosis, cardiovascular diseases, Alzheimer's disease, cancer and so on [4–8]. Up to now, a variety of methods for HClO and MPO detection have been developed, such as chemiluminescent [9,10], colorimetric [11,12], electrochemical [13–16] and fluorescent/phosphorescent methods [17–20], amongst others. However, a limited detection sensitivity is still faced by most of them, and developing a new method for the highly sensitive quantification of HClO and MPO is still urgently needed.

Recently, it has been reported that DNAs with phosphorothioate modification are unstable and sensitive to oxidation by HClO, resulting in DNA strands breaking at the

phosphorothioate site. The underlying mechanism has been comprehensively explained by the Dedon group [21]. This finding makes it possible to develop highly sensitive and specific detection methods for HClO and MPO detection by utilizing the powerful signal-amplification capability of various forms of DNA amplification techniques. By embedding phosphorothioate modification in a DNAzyme, Xiang's group constructed a HClO-activatable $Zn^{2+}$ DNAzyme, demonstrating the feasibility of HClO detection using DNA biosensors for the first time [22]. Through comprehensively studying the DNA cleavage ability of HClO, the Zhang group found that HClO produced by the MPO-$H_2O_2$-$Cl^-$ system could work on the phosphorothioate sites in both single-stranded and double-stranded DNA [23]. They named this HClO-dependent DNA cleavage behavior "DNA scissors" and thus developed an MPO biosensor based on the hybridization chain reaction (HCR), providing a detection limit of 8.35 ng/mL. Furthermore, our group used HClO to trigger DNA-tetrahedron-mediated hyperbranched HCR [24,25], realizing a further improvement of detection sensitivities (0.8 nM for HClO and 3.75 ng/mL for MPO). All these methods are developed on the basis of enzyme-free DNA amplification techniques. Since enzyme-catalyzed techniques usually show a higher signal-amplification capability than enzyme-free ones, developing DNA-based biosensors using enzyme-catalyzed signal-amplification reactions, such as polymerase chain reaction (PCR), recombinase polymerase amplification (RPA) and rolling circle amplification (RCA), may realize the quantification of HClO and MPO with higher sensitivities.

As an enzyme-catalyzed DNA amplification technique, RCA is performed in an isothermal mode, eliminating the requirement for special instruments to achieve rapid temperature changes, which are needed by PCR [26–28]. This characteristic makes RCA a very promising technique for biosensing applications. Herein, by utilizing HClO-induced phosphorothioate site-specific DNA cleavage to trigger an RCA reaction, an RCA-assisted biosensing strategy has been developed, which can not only perform the sensitive and specific detection of HClO and MPO but also work well to screen MPO inhibitors and evaluate their inhibitory effects, showing promising applications in disease diagnosis and drug screening.

## 2. Materials and Methods

### 2.1. Oligonucleotides

All DNA oligonucleotides were synthesized by Sangon Biotech. Co. Ltd. (Shanghai, China), and the sequences are listed in Table 1. Other materials and reagents are given in the supporting information file.

**Table 1.** The oligonucleotides used in this work.

| Oligonucleotide | Sequence (5′→3′) |
|---|---|
| Padlock | P-TGACATTGTACCTCAGCCCTAACCCTAACCCTAACCCTTACCC TAACCCTAACCCTAACCCTCAGCTTGAATCCGTG |
| Primer-S | TACAATGTCACACGGATTCACGTG*TGACATTGTA |
| Primer (mimic cleavage product) | TACAATGTCACACGGATTCACGTG |
| FAM-Primer-S | FAM-TACAATGTCACACGGATTCACGTG*TGACATTGTA |
| FAM-Primer | FAM-TACAATGTCACACGGATTCACGTG |

"P" indicates phosphate group; "*" indicates phosphorothioate-modified site.

### 2.2. HClO Quantitation

A 20 μL detection system containing different concentrations of HClO, 1 μL Primer-S (10 μM, prepared in advance through incubation at 95 °C for 5 min and then at 25 °C for 30 min) and 2 μL 10 × hybridization buffer (500 mM NaCl, 100 mM Tris-HCl, 100 mM $MgCl_2$, 1 mg/mL BSA, pH 7.9) was prepared. The mixture was incubated at 37 °C for 20 min to perform site-specific cleavage of Primer-S by HClO. Then, 4 μL 10 × T4 DNA ligase reaction buffer, 1.5 μL Padlock (10 μM), 20 U T4 DNA ligase and 30.5 μL DEPC water were

added to the detection system. The mixture was incubated at 16 °C for 2 h to perform the cyclization reaction of Padlock.

A total of 4 μL 10× phi29 DNA polymerase reaction buffer, 5 μL thioflavin T (ThT) (200 μM), 2.5 μL dNTPs (10 mM), 6 U phi29 DNA polymerase and water were added to the detection system to a final volume of 100 μL. The mixture was incubated at 30 °C for 30 min for the RCA reaction. Then, the fluorescence spectrum was recorded using 425 nm as the excitation wavelength, and the fluorescence signal at 485 nm was used for HClO quantitation.

### 2.3. MPO Quantitation

MPO was diluted in 1 × MPO buffer (5 mM NaAc-HAc, pH 6.0 with 10 mM NaCl). A 20 μL detection system containing different concentrations of MPO, 1 μL $H_2O_2$ (200 μM), 1 μL Primer-S (10 μM, prepared in advance through incubation at 95 °C for 5 min and then at 25 °C for 30 min) and 2 μL 10 × hybridization buffer (500 mM NaCl, 100 mM Tris-HCl, 100 mM $MgCl_2$, 1 mg/mL BSA, pH 7.9) was prepared. The mixture was incubated at 37 °C for 20 min to perform MPO-catalyzed HClO production and site-specific cleavage of Primer-S. Then, the cyclization of Padlock, RCA reaction, and fluorescence signal detection were the same as those for HClO detection.

## 3. Results

### 3.1. Working Mechanism of RCA-Assisted Biosensor for HClO Detection

Using HClO-induced phosphorothioate site-specific cleavage of DNA to trigger RCA, a DNA-assisted biosensing platform is developed for HClO detection (Scheme 1). Only two DNA oligonucleotides (Primer-S and Padlock), neither fluorescently labelled, are used. Different from the linear primer used in traditional RCA, the primer (named as Primer-S) used in this work is designed as a hairpin, and a thiophosphate modification site is embedded in the double-stranded stem of Primer-S. The robust stability of the hairpin structure hinders the hybridization between Primer-S and Padlock, a linear DNA strand that is used to prepare the circular template of RCA. As a result, the circular RCA template cannot be formed, and, thus, the subsequent RCA reaction will not be initiated. In the presence of HClO, however, Primer-S is cleaved into two parts by HClO at the thiophosphate site, resulting in the destruction of the hairpin structure and the release of the loop sequence (yellow part of Primer-S, Scheme 1). The resultant long strand (named Primer) can then hybridize with the Padlock strand using the released loop sequence as the toehold, bridging the 5′-phosphate and 3′-OH ends of Padlock together. In the presence of T4 DNA ligase, the two ends of Padlock are linked, resulting in the cyclization of Padlock. Using the resultant circular Padlock as the template and the long cleavage fragment of Primer-S (Primer) as the primer, the RCA reaction is initiated successfully, producing long-stranded products containing large numbers of tandem repeat sequences. If a well-designed C-rich sequence is embedded in the middle of Padlock (black part of Padlock in Scheme 1), RCA products containing G-rich tandem repeats will be produced. The G-rich repeats can then fold into large amounts of G-quadruplexes, which are able to specifically bind to thioflavin T (ThT) to produce a fluorescent signal [29], realizing the label-free detection of HClO.

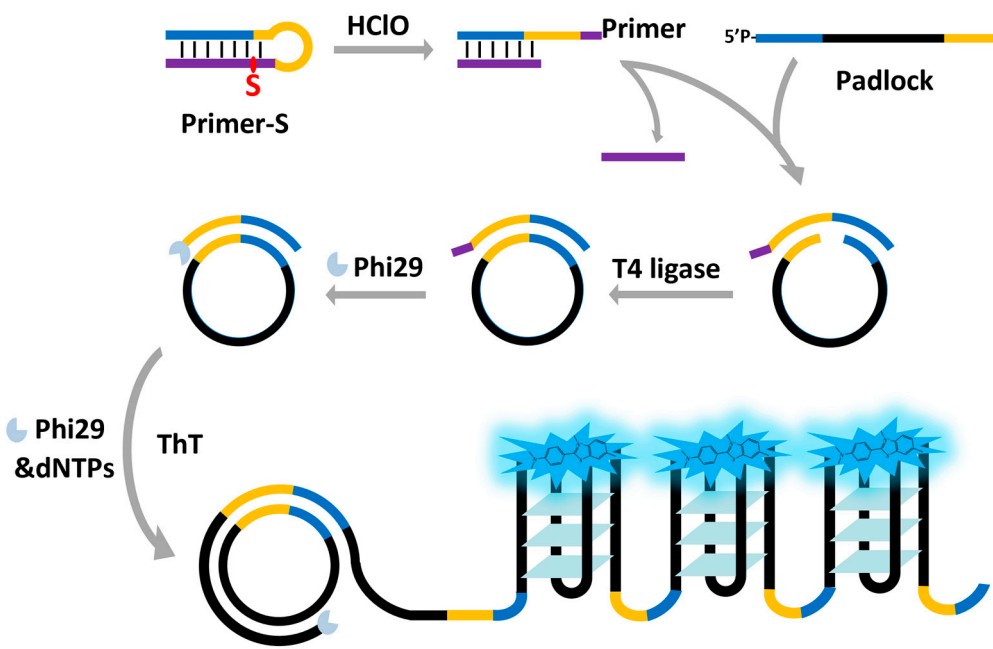

**Scheme 1.** Working mechanism of RCA-assisted biosensor for HClO detection.

### 3.2. Feasibility Verification of HClO-Sensing Strategy

Polyacrylamide gel electrophoresis (PAGE) and fluorescence analysis were carried out to verify the feasibility of the proposed sensing strategy. As shown in Figure 1a, treatment of Primer-S with HClO resulted in the emergence of a new band (Lane 2), whose migration rate was faster than that of intact Primer-S (Lane 1) but similar to that of simulated long-strand cleavage product Primer (Lane 3), illustrating that Primer-S had been precisely cleaved by HClO at the thiophosphate site. Having demonstrated the site-specific cleavage of Primer-S by HClO, we investigated the effects of HClO on the interactions between Primer-S and Padlock. As shown in Figure 1b, without HClO, intact Primer-S kept the hairpin structure rather than hybridizing with Padlock. Their mixture showed two distinct bands (Lane 3), corresponding with Primer-S (Lane 1) and Padlock (Lane 2), respectively. The addition of HClO in the Primer-S/Padlock mixture resulted in the emergence of the third band (Lane 4), whose migration rate was slower than those of both Primer-S and Padlock and identical to that of the mixture of Padlock and simulated product Primer (Lane 5). These results are perfectly consistent with the proposed working mechanism; that is, Primer-S has been cleaved by HClO, and the resultant long fragment hybridizes with Padlock to form the Primer/Padlock complex.

Under the action of T4 DNA ligase and Phi29 DNA polymerase, the cleaved Primer-S could induce the cyclization of Padlock and thus initiate RCA reaction, producing long-stranded amplification products with tandem G-rich sequences. Thus, one could see that a bright band was barricaded at the sample-adding port in the PAGE assay (Figure 1c, Lane 2). When ThT was added in the reaction system, a significantly increased fluorescence was observed due to the fluorescent response of ThT to the G-quadruplexes formed by G-rich RCA products (Figure 1d). A similar bright band (Figure 1c, Lane 1) and fluorescence enhancement were observed when simulated long-stranded cleavage product Primer was added in place of Primer-S and HClO. By comparison, no amplification product or enhanced fluorescence was observed in the absence of HClO, T4 DNA ligase, Primer-S, Padlock or Phi29 DNA polymerase (Figure 1c,d). The above experimental results jointly demonstrated the feasibility of our RCA-assisted strategy for HClO detection.

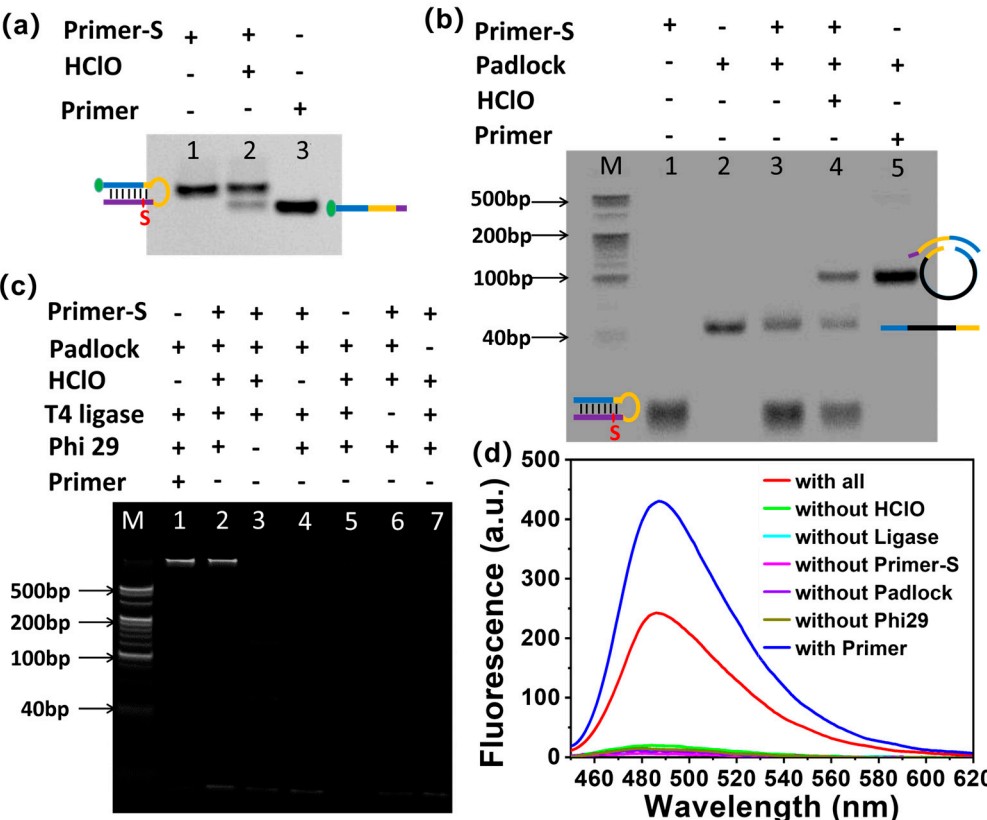

**Figure 1.** Feasibility verification of RCA-assisted biosensor for HClO detection. (**a**) A 24% denaturing of the site-specific cleavage of Primer-S by HClO. Fluorophore FAM is labelled at the 5′-end of DNA to clearly show the migration of DNA band. Primer is the simulated long-stranded cleavage product of Primer-S. (**b**) A 10% non-denaturing PAGE assay of the hybridization between Primer-S and Padlock in the absence or presence of HClO. (**c**) A 10% non-denaturing PAGE assay of HClO-initiated RCA reaction. (**d**) Fluorescence analysis of the proposed HClO-sensing system. [HClO] = 3 μM.

### 3.3. Sensitivity and Specificity of HClO Biosensor

Before evaluating the sensitivity of the RCA-assisted biosensor, several vital experimental conditions were optimized to obtain the best HClO-sensing performance, including the concentrations of Primer-S, Padlock, and Phi29 DNA polymerase and the reaction time for HClO-dependent Primer-S cleavage and the subsequent RCA reaction (Figures S1–S5). Under the optimized conditions (100 nM Primer-S, 150 nM Padlock, 6 U Phi29 DNA polymerase, 20 min for Primer-S cleavage and 30 min for RCA reaction), the detection sensitivity of the proposed HClO-sensing strategy was investigated based on the fluorescence at 485 nm produced by the specific fluorescent response of ThT to G-quadruplexes. As shown in Figure 2a,b, the fluorescence intensity was positively correlated with the HClO concentration, as a result of which the more HClO the sensing system contained, the more Primer-S strands were cleaved to trigger the RCA reaction and the more G-quadruplexes were produced. In the range of 5 nM~5 μM, the fluorescence intensity was linear with the concentration of HClO (Figure 2c), with a linear relationship ($R^2 = 0.9915$) of $F = 76.31 C_{HClO}$ (μM) + 14.45 ($F$ was the fluorescence intensity at 485 nm, and $C_{HClO}$ was the HClO concentration). This RCA-assisted HClO biosensor showed a good sensitivity, with the limit of detection (LOD) calculated as 1.67 nM, based on the $3\sigma/S$ rule, which is superior to or equivalent to other fluorescent detection methods of HClO (Table S1) [11,19,24,30–32].

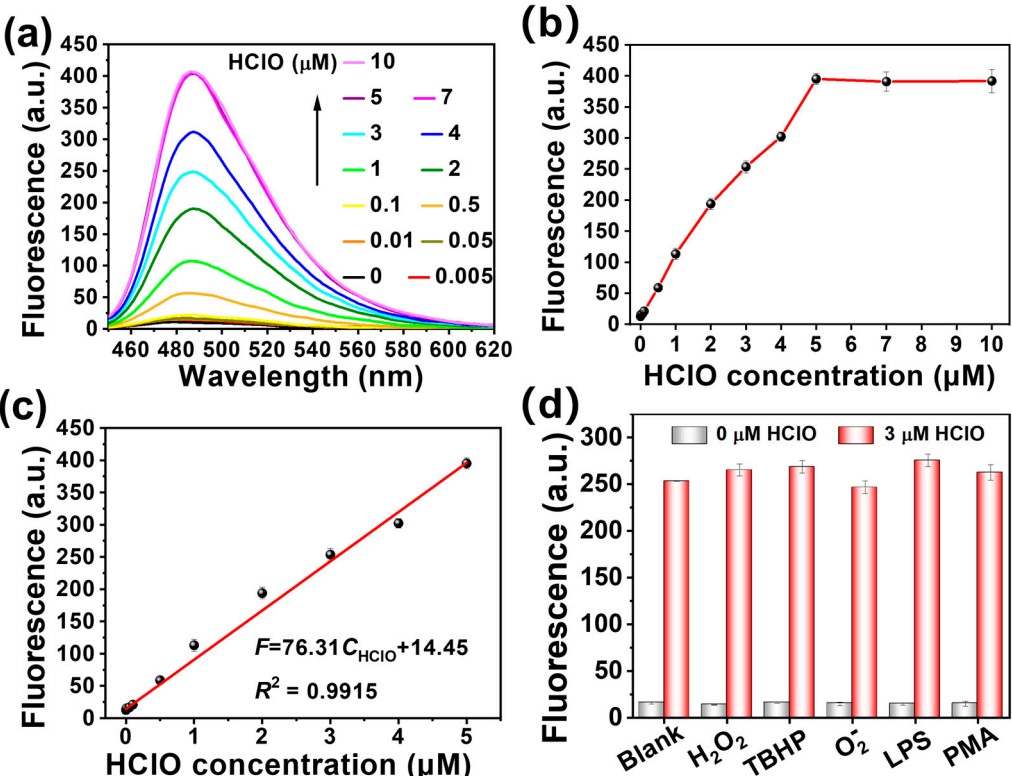

**Figure 2.** Sensitivity and selectivity of the HClO-sensing system. (**a**) Fluorescence spectra of the sensing systems containing different concentrations of HClO. (**b**) Fluorescence intensity at 485 nm as a function of HClO concentration. (**c**) The linear relationship between fluorescence intensity at 485 nm and HClO concentration in the range of 5 nM~5 μM. (**d**) Selectivity of the HClO-sensing system. The black bars represent the fluorescence signals of the sensing systems containing other ROS or chemicals. The red bars represent the fluorescence signals of the sensing systems containing other ROS (or chemicals) and 3 μM HClO. [$H_2O_2$] = [TBHP] = [$O_2^-$] = 100 μM. [LPS] = [PMA] = 1 μg/mL.

Furthermore, the specificity and anti-interference capability were investigated to further evaluate the performance of this RCA-assisted HClO biosensor. Its fluorescent responses to several other ROS ($H_2O_2$, $O_2^-$, tert-butyl hydroperoxide (TBHP)) and the chemicals (e.g., lipopolysaccharide (LPS) and phorbol 12-myristate 13-acetate (PMA)) that can induce intracellular HClO production were examined [33,34]. The results showed that their presence could neither cause a detectable fluorescent response, nor interfere with the quantification of HClO (Figure 2d), reflecting the good detection specificity and strong anti-interference ability of this HClO-sensing system.

### 3.4. HClO Detection in Complex Biological Sample

In order to further investigate the practical application potential of this biosensor, the recovery of HClO in 100-fold-diluted healthy human serum was tested. As shown in Table S2, the recoveries were in the range of 93.43–102.17%, suggesting that the proposed RCA-assisted HClO biosensor holds great promise for the detection of HClO in real samples.

### 3.5. Working Mechanism and Feasibility Verification of Derivative MPO Biosensor

The above proposed sensing strategy shows an excellent HClO detection specificity. Thus, it can be reasonably speculated that this strategy may also be utilized to monitor the HClO-producing reaction. As far as we know, under physiological conditions, MPO can catalyze the redox reaction between $H_2O_2$ and $Cl^-$ to generate HClO. When the concentrations of $H_2O_2$ and $Cl^-$ remain constant, the amount of produced HClO is positively

correlated with the MPO concentration. Therefore, the RCA-assisted HClO-sensing strategy can also be used for the quantification of MPO activity (Figure 3a).

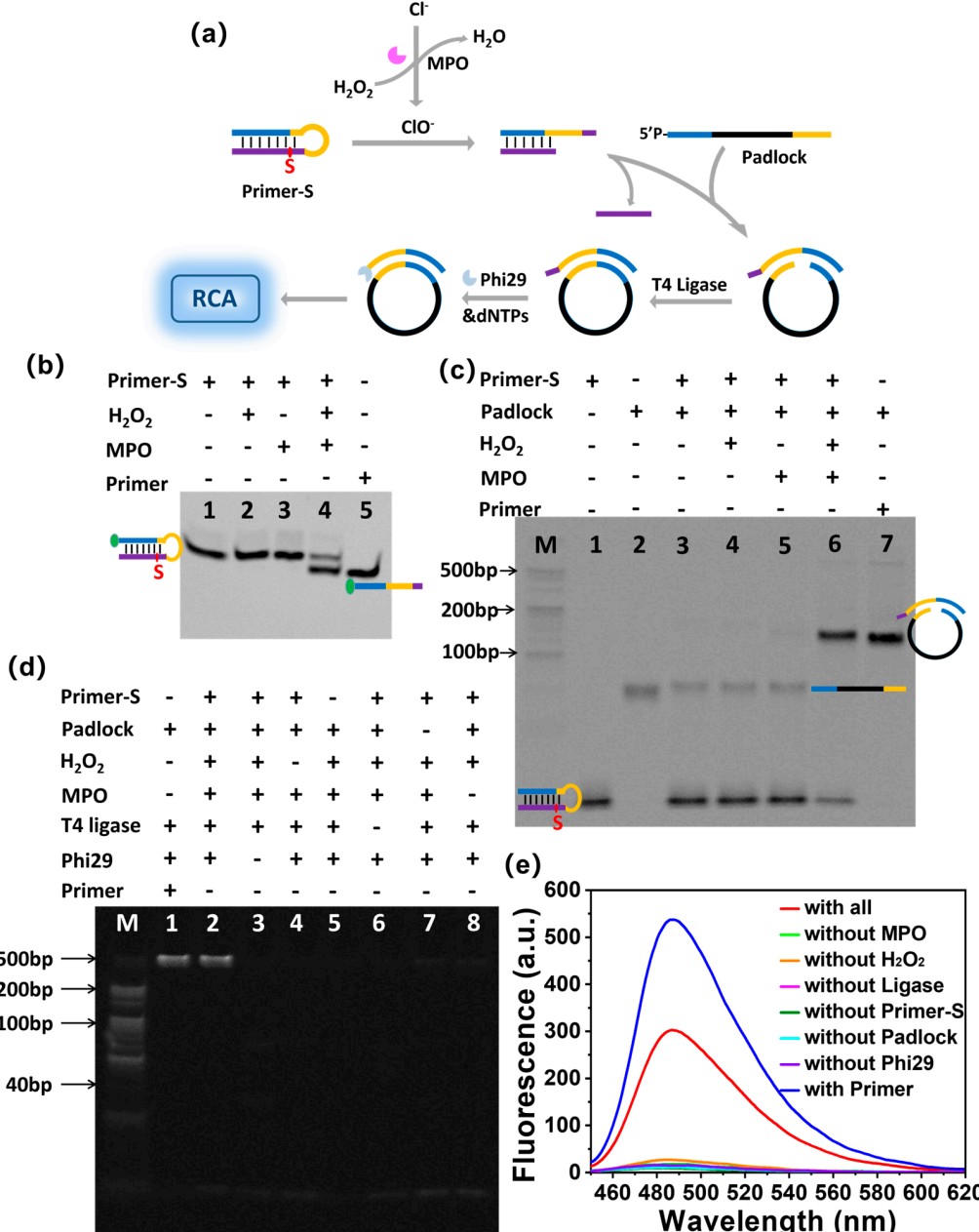

**Figure 3.** Working mechanism and feasibility verification of RCA-assisted MPO sensor. (**a**) Working mechanism of the proposed RCA-assisted MPO-sensing system. (**b**) A 24% denaturing PAGE assay of the site-specific cleavage of Primer-S by MPO-$H_2O_2$-Cl$^-$ system. Fluorophore FAM is labelled at the 5′-end of DNA to clearly show the migration DNA band. (**c**) A 10% non-denaturing PAGE assay of the hybridization between Primer-S and Padlock in the absence or presence of MPO. (**d**) A 10% non-denaturing PAGE assay of MPO-initiated RCA reaction. (**e**) Fluorescence analysis of the proposed MPO-sensing strategy.

Such a speculation has been confirmed by a series of verification experiments. The results of PAGE assays showed that Primer-S could only be broken precisely at the thiophosphate site when all three components (MPO, $H_2O_2$ and Cl$^-$) existed (Lane 4, Figure 3b). The resultant cleavage product then hybridized with linear Padlock (Lane 6, Figure 3c) to prepare a circular template for subsequent RCA, producing long-stranded amplification

products (Lane 2, Figure 3d) containing tandem repeats of the G-rich sequence, which folded into G-quadruplexes and was specifically recognized by fluorescent probe ThT (Figure 3e). In contrast, these successive reactions did not occur with a lack of either $H_2O_2$ or MPO. Therefore, if $H_2O_2$ and $Cl^-$ are pre-existing in the system, MPO can specifically initiate the RCA reaction, providing a greatly increased fluorescent signal output and thus achieving the quantification of MPO activity.

### 3.6. Sensitivity and Specificity of MPO Biosensor

Based on the optimal conditions of the aforementioned HClO detection, the concentration of $H_2O_2$ and the cleavage time of Primer-S by the MPO-$H_2O_2$-$Cl^-$ system were further optimized for MPO detection (Figures S6 and S7). Ultimately, the following conditions were selected: 100 nM Primer-S, 150 nM Padlock, 10 μM $H_2O_2$, 6 U Phi29 DNA polymerase, 15 min for Primer-S cleavage by MPO-$H_2O_2$-$Cl^-$ system, and 30 min for RCA reaction. Excess $Cl^-$ was provided by the reaction buffers. Under the optimized conditions, the MPO detection sensitivity of the proposed method was investigated. As shown in Figure 4a,b, the fluorescence intensity of the sensing system increased monotonically with the MPO concentration, and a linear relationship ($R^2 = 0.9857$) was obtained in the MPO concentration range of 1~400 ng/mL, providing a linear regression equation of $F = 1.04\ C_{MPO}(ng/mL) + 15.23$ ($F$ is the fluorescence value at 485 nm, and $C_{MPO}$ is the MPO concentration). Based on the $3\sigma/S$ rule, the LOD was calculated to be 0.33 ng/mL, which is comparable to or better than reported fluorescent sensors of MPO (Table S3) [19,20,23,24,32,35,36].

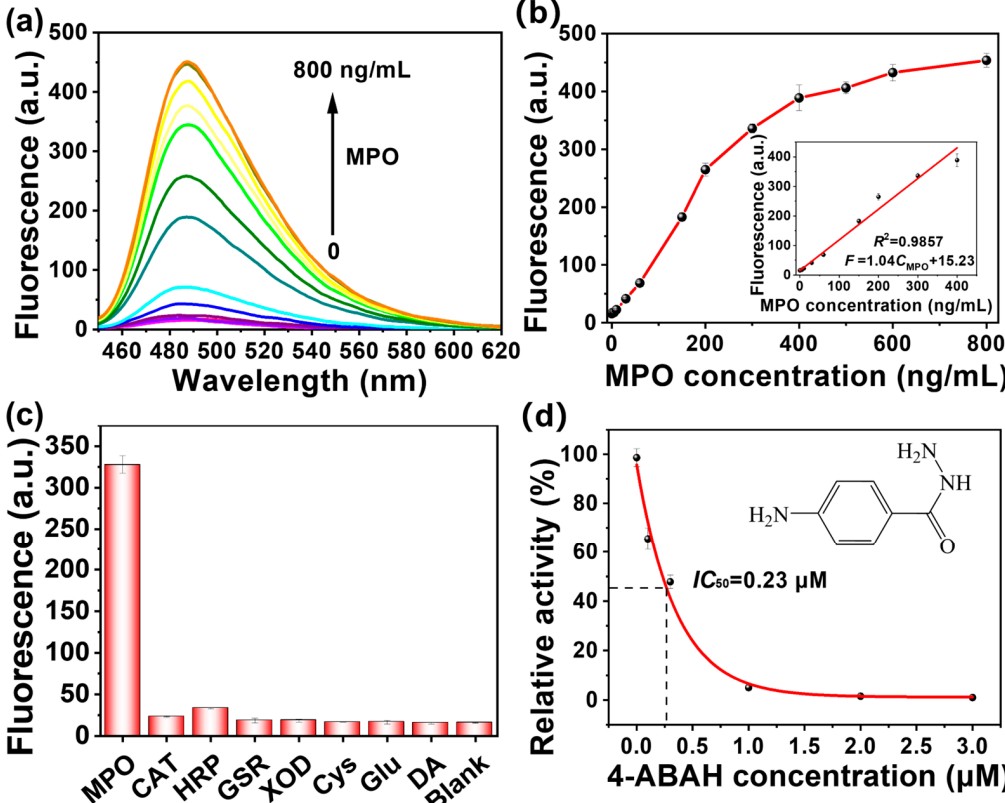

**Figure 4.** Sensitivity, selectivity and inhibitor screening application of the MPO-sensing system. (**a**) Fluorescence spectra of the sensing systems containing different concentrations of MPO. (**b**) Fluorescence intensity at 485 nm as a function of MPO concentration. The insert shows the linear relationship between fluorescence intensity and MPO concentration in the range of 1~400 ng/mL. (**c**) Selectivity of the MPO-sensing system. [MPO] = [Cys] = [Glu] = [DA] = 300 ng/mL, [CAT] = [HRP] = [GSR] = [XOD] = 0.01 U/μL. (**d**) Relative activity of MPO in the presence of different concentrations of 4-ABAH.

Moreover, this sensing system showed a high specificity for MPO detection. When several kinds of redox-related enzymes and common biomolecules, such as catalase (CAT), horseradish peroxidase (HRP), glutathione reductase (GSR), xanthione oxidase (XOD), L-cysteine (Cys), glucose (Glu) and dopamine (DA), were added in the sensing system instead of MPO, none of them was able to cause an obvious fluorescent response compared to the blank control (Figure 4c), suggesting that the sensing system can only be activated by MPO.

*3.7. MPO Detection in Complex Biological Sample*

MPO is becoming an important biomarker for the diagnosis of ROS-related diseases. This biosensor, capable of MPO detection in complex biological samples, will hold great promise in practical applications. Therefore, the application feasibility of the proposed MPO biosensor in human serum was investigated. The recoveries of different concentration levels of MPO from 100-fold-diluted healthy human serum were determined to be in the range of 94.59–107.39%, and the detection results were consistent with those of the commercial kit (Table S4), illustrating the practical application potential of this MPO biosensor in complex biological samples.

*3.8. Application in MPO Inhibitor Analysis*

Besides being a disease biomarker, MPO also hold great promise as a therapeutic target. Thus, MPO inhibitors, which are able to inhibit the activity of MPO, may be developed as MPO-targeted drugs. Thus, the inhibitor screening ability of the RCA-assisted MPO biosensor was evaluated using 4-aminobenzoic acid hydrazide (4-ABAH) as the model inhibitor [37]. Pre-incubation of MPO (300 ng/mL) with 4-ABAH before an activity assay caused a significant decrease in the fluorescent signal output of the sensing system, and the fluorescence decrease was 4-ABAH-concentration-dependent, reflecting the activity inhibition of MPO by 4-ABAH (Figure 4d). By following the 4-ABAH-concentration-dependent MPO activity changes, which were calculated from the 4-ABAH-induced fluorescence decrease of the MPO-sensing system, the half inhibitory concentration ($IC_{50}$) of 4-ABAH was determined to be 0.23 μM, which is consistent with previous reports [38]. These results suggest that our MPO biosensor can be used not only to screen MPO inhibitors, but also to evaluate the inhibitory capability of the inhibitors, thus showing great application potential in drug analysis.

**4. Conclusions**

In summary, by utilizing HClO-induced highly specific cleavage of thiophosphate-modified hairpin DNA to trigger an isothermal RCA reaction, an RCA-assisted HClO biosensor was developed. The powerful signal amplification capability of RCA endows the proposed biosensor with a high detection sensitivity, and the specific response of ThT to G-quadruplexes makes a label-free fluorescence detection possible. The as-proposed HClO biosensor was demonstrated to work well for a sensitive and specific HClO detection with a LOD of 1.67 nM. Moreover, by utilizing the MPO-catalyzed $H_2O_2$-$Cl^-$ reaction to produce HClO, the HClO-sensing system can be further developed as an MPO biosensor, which works well not only for the sensitive and specific detection of MPO (LOD: 0.33 ng/mL), but also for the screening and activity evaluation of MPO inhibitors, thus showing great application potentials in disease diagnosis and drug screening.

**Supplementary Materials:** The following supporting information can be downloaded at: https://www.mdpi.com/article/10.3390/chemistry5020098/s1, Figure S1: Optimization of Primer-S concentration; Figure S2: Optimization of Padlock concentration; Figure S3: Optimization of Phi29 DNA polymerase amount; Figure S4: Optimization of the reaction time for HClO-induced Primer-S cleavage; Figure S5: Optimization of RCA reaction time; Figure S6: Optimization of $H_2O_2$ concentration; Figure S7: Optimization of the reaction time for MPO-dependent Primer-S cleavage; Table S1: Comparison of several HClO detection methods; Table S2: Recovery of HClO from human serum

sample; Table S3: Comparison of several MPO detection methods; Table S4: Recovery of MPO from human serum sample.

**Author Contributions:** Conceptualization, data curation, formal analysis, investigation, writing original draft, B.L.; conceptualization, investigation, methodology, writing original draft, J.-Y.M.; investigation, J.W.; visualization, D.-X.W.; supervision, investigation, A.-N.T.; supervision, funding acquisition, writing review & editing, D.-M.K. All authors have read and agreed to the published version of the manuscript.

**Funding:** This research was funded by the National Key R&D Program of China (2019YFA0210103) and the National Natural Science Foundation of China (Nos. 22074068).

**Data Availability Statement:** Not applicable.

**Conflicts of Interest:** The authors declare no conflict of interest.

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
