# Peer review of "Rolling-Circle-Amplification-Assisted DNA Biosensors for Sensitive and Specific Detection of Hypochlorous Acid and Myeloperoxidase"

_chemistry, doi:10.3390/chemistry5020098_

Round 1

Reviewer 1 Report

In this work, the authors reported a highly sensitive and specific biosensing strategy for the quantitation of hypochlorous acid (HClO) and myeloperoxidase (MPO). The powerful signal amplification capability of RCA endows the sensing systems with high sensitivity. The specific fluorescent response of thioflavin T (ThT) to G-quadruplexes in RCA products makes label-free signal output possible. The detection limits for HClO and MPO reached 1.67 nM and 0.33 ng/mL, respectively. Overall, the idea is interesting, and the results are important. However, there are some issues that need to be addressed.

(1) The MPO catalytic reaction requires the participation of chloride ions. There is no evaluation of the effect of the reaction system on the concentration of chloride ions. Why?

(2) In the section “3.3. Sensitivity and specificity of HClO biosensor”, why does the fluorescence signal decrease as the concentration of Padlock increases in Figure S2?

(3) In the section “3.3. Sensitivity and specificity of HClO biosensor”, why choose LPS and PMA when determining the specificity of the sensing system?

(4) The experimental details of RCA reaction are missing in the 2.2 section.

(5) Please check Figure 1d and Figure 3e. I think Pih29 should be Phi29.

(6) Some physical quantities such as F, C, S, and IC50 should be italic, as shown in Figure 2c.

(7) Corresponding references concerning the specific fluorescence response of ThT to G-quadruplexes should be given.

Minor editing of English language is required.

Reviewer 2 Report

Liu and Ma reported an HClO sensing approach using site-specific cleavage of phosphorothioate-modified DNA and sequential rolling circle amplification (RCA). This method showed a low detection limit and some selectivity. The sensing in complex biological samples of human serum suggested the potential application in real samples. The work is interesting to a broad audience in the fields of fluorescence sensing and RCA-assisted biosensor designs. It is recommended for publication after minor revision.

1. the concentrations of HClO in Figure 1 should be included in the figure caption.

2. the sensing procedure involves several steps, including cleavage of modified DNA, RCA, and the introduction of ThT. It is recommended to add some discussion about this, for example, which step is most critical and how this approach is in comparison with other HClO sensing methods.

3.  It would be interesting to see how HClO4 would react to the modified DNA and affect the selectivity. 

Minor editing of the English language 
